# Influencing Factors on Person-Centered Care Competence among Nursing Students Experienced Clinical Training

**DOI:** 10.3390/medicina57121295

**Published:** 2021-11-25

**Authors:** Juhyun Ahn, Myoungsuk Kim

**Affiliations:** Department of Nursing, College of Nursing, Kangwon National University, Chuncheon-si 24341, Korea; saeucap@gmail.com

**Keywords:** person-centered care, nursing professional value, satisfaction with major, nursing student

## Abstract

*Background and Objectives*: Nursing students require appropriate education to improve their person-centered care competence. Therefore, research on the various factors that influence person-centered care competence is necessary. This study aimed to identify factors influencing nursing professional values, satisfaction with major, and perception of the nursing profession on person-centered care competence. *Materials and Methods*: This study was a descriptive survey, and participants were nursing students from three universities in Korea. Structured self-report questionnaires were used for data collection. *Results*: Nursing professional values (*p* < 0.001) were found to be an influencing factor on person-centered care competence (Adjusted R^2^ = 0.244). However, the perception of the nursing profession, and the satisfaction with the major were not found to be significant influencing factors on person-centered care competence. *Conclusions*: The findings suggest that fostering nursing professional values in nursing students and developing educational interventions for the same are essential to improve person-centered care competence.

## 1. Introduction

Person-centered care (PCC) is a holistic approach involving respecting and individualizing each patient, facilitating their autonomy to the greatest possible extent, and allowing them to actively participate in their treatment [1]. PCC is also used interchangeably with patient-centered care, both focusing on empathy, communication, respect, and decision-making. However, they differ in that the goal of patient-centered care is to achieve a functional life for the patients through disease treatment, while that of PCC is to achieve a meaningful life for the patients [2]. PCC offers positive results for both patients and healthcare providers by positively impacting patients’ lifestyles, reducing the risk of contracting chronic diseases, extending life expectancy [3,4], reducing the risk of falls, decreasing behavioral-psychological symptoms in patients with dementia, and improving health-related quality of life, well-being, and safety [5,6]. In addition, the positive effects for healthcare providers include improved job satisfaction and lower stress [7,8].

Globally, PCC has been developed not only in theory but also in practice through research [9], and many studies have evaluated the effects of PCC implementation in the clinical or long-term care setting [3,10,11,12,13]. However, most of these studies focused on PCC education for nurses rather than nursing students [14]. Furthermore, there is a lack of clarity regarding the PCC definition and the ideal education and evaluation methods for nursing curricula [15]. PCC aims to provide humanistic care that transcends the mere focus on an individual’s disease; it begins with the process of recognizing patients as human beings [2]. It is a relatively advanced form of nursing, and therefore, a specialized, incremental approach for its education is required in nursing curricula [16]. Additionally, for nurses to provide systematic PCC that accords with contemporary trends and healthcare and medicine requirements, PCC competence must be promoted in the nursing curriculum [7]. Therefore, such curricula should include various educational programs to promote PCC competence by identifying factors affecting it.

According to existing literature, several related factors that influence PCC competence in nursing students have been identified, including professional nursing competence [17], interpersonal skills [18,19], empathic competence [17,18,20], and perceived stress [18]. However, no studies have been reported on the influencing factors of nursing professional values, satisfaction with one’s major, and the perception of the nursing profession on PCC competence.

First of all, professional values are behavioral principles that provide basic standards for planning and judging aims and actions [10]; thus, it may be argued that nursing professional values are essential to the practice of professional nursing. Nursing professional values are significant predictors of quality of care and job satisfaction; therefore, nursing students need to acquire appropriate nursing professional values [21]. In addition, nursing professional values in clinical nurses have been reported to be the most influential factor on PCC competence [22], and the resolute and correct establishment of nursing professional values among nursing students in curriculum is important to increase PCC competence [23].

Further, satisfaction with one’s major among nursing students can be an important factor in job satisfaction and willingness to continue one’s career after graduation [24]. Previous studies [18,19] have reported that satisfaction with one’s college major influences PCC competence. However, these researchers were incapable of measuring this factor because they only allocated a single question related to satisfaction to one’s major. Therefore, to confirm whether nursing students’ satisfaction with their major affects PCC competence, we require accurate and comprehensive research. Among nursing students, satisfaction with major is an important factor that forms a positive value on nursing, affects participation in clinical training and education, and also affects college life adaptation [25]. Therefore, a study is needed to identify the effect of nursing students’ satisfaction with their major on PCC competence.

Finally, perception of the nursing profession is a concept that includes one’s ideas, beliefs, impressions, and perspectives of nursing [26]. The decision to enter a nursing department and pursue a career in nursing may be influenced by students’ perceptions of the profession [27]. Thus, encouraging nursing students to develop a positive perception of the nursing profession can help them foster a professional identity as nurses, which further has a positive effect on their roles as nurses [28]. As such, perception of the nursing profession is essential for nursing students who will become nurses, so it is necessary to identify whether it is a factor influencing PCC competence.

No study has yet to identify the influencing factors of nursing professional values, satisfaction with one’s major, and the perception of the nursing profession on PCC competence. In addition, various factors affecting PCC competence should be identified to improve PCC competence among nursing students. Therefore, this study aims to contribute to the development of programs to improve PCC competence by identifying factors that affect PCC competence, and to obtain data to promote nursing students’ PCC competence.

## 2. Methods

### 2.1. Study Design

This study is a descriptive survey that identifies the impacts of nursing professional values, satisfaction with the major, and perception of the nursing profession on the nursing students’ PCC competence.

### 2.2. Participants

This study involves those nursing students as participants who (1) attended one of three nursing colleges located in Kangwon Province, Korea, (2) had received at least six months of clinical training, and (3) had voluntarily agreed to participate. The sample size was determined based on consideration of a previous study of factors that potentially influence nursing students’ PCC competence [18], which means a medium effect size (0.15) was used, and nine predictors were selected. Using the G*power 3.1.9 software (Heinrich-Heine-University, Düsseldorf, Germany), we determined that for a significance level of 0.05, a medium effect size of 0.15, and a power of 0.9, the minimum sample size was 152 persons. Considering potential dropouts and eliminations, we distributed 170 questionnaires. All 170 questionnaires were collected, of which 12 featured insincere questionnaires with incomplete responses. Ultimately, 158 appropriate questionnaires were analyzed.

### 2.3. Instruments

This study used four measurement tools as instruments to measure the variables. Participants’ demographic data were also collected, including age, gender, grade, religion, satisfaction with clinical training, and academic score (4.5 point) in the preceding semester.

#### 2.3.1. Person-Centered Care Competence

An instrument developed by Suhonen et al. [29], which was adapted into Korean by Park [20], was used to measure the participants’ PCC competence. This instrument comprises 17 questions, each of which is scored using a 5-point Likert scale. Higher scores indicate higher PCC competence. For Park [20], the reliability of the instruments by Cronbach’s α was 0.89; for the present study, it was 0.85.

#### 2.3.2. Nursing Professional Values

An instrument developed by Yeun, Kwon, and Ahn [23] for nurses and modified by Han et al. [30] for nursing students, was used to measure the participants’ nursing professional values. This instrument comprises 18 questions, each of which is scored using a 5-point Likert scale. Higher scores indicate higher nursing professional values. For Han, Kim, and Yun [30], the reliability of the instruments by Cronbach’s α was 0.91; for the present study, it was 0.80.

#### 2.3.3. Satisfaction with Major

An instrument developed by Ha [31] and modified by Lee [32] was used to measure the participants’ satisfaction with their major. This instrument comprises 18 questions, each of which is scored using a 5-point Likert scale. Higher scores indicate higher satisfaction with the major. For Lee [32], the reliability of the instruments by Cronbach’s α was 0.90; for the present study, it was 0.89.

#### 2.3.4. Perception of the Nursing Profession

An instrument developed by Kang, Go, Yang and Kim [26] was used to measure participants’ perception of the nursing profession. This tool comprises 20 questions, each of which is scored using a 5-point Likert scale. Higher scores indicate a more positive perception of the nursing profession. For Kang, Go, Yang, and Kim [26], the reliability of the instruments by Cronbach’s α was 0.94; for the present study, it was 0.90.

### 2.4. Data Collection

Data collection was performed from 1 December 2020 to 31 January 2021, at three nursing colleges located in Kangwon Province. The research assistant placed a research recruitment notice on the nursing departments’ respective bulletin boards. Potential subjects then voluntarily applied to participate in the research, signed consent forms, and sent them to the research assistant. After completing the initial formalities, subjects were allowed to respond to the online questionnaire. A small gift coupon was provided to each participant upon completion of the questionnaire.

### 2.5. Ethical Considerations

This study was approved by the Institutional Review Board of Kangwon National University (IRB No. 2020-07-004-001) before the commencement of data collection and was conducted following the stipulations of the Declaration of Helsinki. Before responding to the online questionnaire, the participants were informed of the purpose of the study and the data-collection procedure and were ensured that their anonymity and confidentiality would be guaranteed, their privacy would be protected, how the research data would be disposed of, and the fact that they could withdraw from the study at any time. The participants voluntarily participated after the submission of written informed consent to participate.

### 2.6. Statistical Analysis

The collected data were analyzed using IBM SPSS Statistics ver. 25.0 software (IBM Corp., Armonk, NY, USA). The participants’ general characteristics, PCC competence, nursing professional values, satisfaction with major, and perception of the nursing profession were analyzed using descriptive statistics, such as frequency with percentage and mean with standard deviation. The differences in PCC competence by participants’ general characteristics were analyzed using independent *t*-tests and one-way analyses of variance. The factors influencing nursing professional values, satisfaction with major, and perception of the nursing profession on PCC competence was analyzed using stepwise multiple linear regression analysis. A *p*-value less than 0.05 was regarded as statistically significant. In addition, the reliability of the instruments was analyzed using Cronbach’s α.

## 3. Results

### 3.1. PCC Competence According to General Characteristics

The participants’ general characteristics are presented in Table 1. Students aged <25 years accounted for 72.2% of the sample, and females accounted for 87.3%. Regarding grade, most of the samples were third-year students (55.1%), while the fourth-year students consisted of 177 participants, accounting for 44.9%. The majority of participants reported having no religion (52.8%). Most subjects (60.1%) had an academic score of 3.0–3.9 (out of 4.5) in the preceding semester, and the majority (66.5%) reported being satisfied with clinical training. In this study, there was no statistically significant difference in the mean of PCC competence for sex (*t* = 1.87, *p* = 0.065), grade (*t* = −0.16, *p* =0.874), religion (*t* = 1.53, *p* = 0.129), satisfaction of clinical practice (*t* = 2.10, *p* = 0.125), and academic score in the preceding semester (*t* = 0.77, *p* = 0.464). On the other hand, PCC competence significantly differed according to age, students aged ≥25 showed higher PCC competence than those aged <25 (*t* = −2.09, *p* = 0.038).

### 3.2. Levels of Person-Centered Care Competence, Nursing Professional Values, Satisfaction with Major, and Perception of the Nursing Profession

The participants’ respective average scores for PCC competence, nursing professional values, satisfaction with the major, and perception of the nursing profession are presented in Table 2. It shows that the average score for PCC competence was 3.69 ± 0.46, for nursing professional values was 3.92 ± 0.40, for satisfaction with the major was 4.10 ± 0.51, and for the perception of the nursing profession was 3.99 ± 0.50 (all scores were based on a 5-point scale).

### 3.3. Factors Influencing Person-Centered Care Competence

The results for the factors influencing the nursing students’ PCC competence are presented in Table 3. We imputed independent variables such as nursing professional values, satisfaction with major, perception of the nursing profession, and age variable, which significantly varied in general characteristics. The nominal variable was dummy-coded. In this study, the regression model was significant (F = 51.61, *p* < 0.001). The range of tolerance was 0.17~0.97 and that of the variance inflation factor (VIF) was less than 10, at 1.02~5.69, demonstrating no risk of multicollinearity. Furthermore, the Durbin-Watson statistic was found to be 1.98, exhibiting no autocorrelation among the error terms. There were no items with a Cook’s distance of ≥1.0, indicating no issues regarding influence. A residual analysis with normal p-p residual plots and scatter plots confirmed the linearity, normality of error terms, and homoscedasticity of the model.

The adjusted R^2^ value for the participants’ PCC competence was 0.244, and the explanatory power of the measurement variables for PCC competence was 24.4%. The factor that showed the most significant influence on the nursing college students’ PCC competence was nursing professional values (*β* = 0.49, *p* < 0.001), which indicated that higher nursing professional values predict higher PCC competence. Additionally, perception of the nursing profession, satisfaction with major, and age did not significantly influence factors on PCC competence.

## 4. Discussion

This study attempted to investigate the respective effects of nursing professional values, satisfaction with major, and perception of the nursing profession on PCC competence to obtain data contributing to developing an effective program for improving nursing students’ PCC competence.

In this study, nursing professional values were identified as an influencing factor on nursing students’ PCC competence. This finding shows the importance of nursing professional values for promoting PCC competence among nursing students. It is similar to a previous result by [22], who finds that nursing professional values influence PCC competence among nurses. Positive and resolute nursing professional values enable the professional performance of one’s role, improve nursing quality, and better application of holistic care [23]. Nursing professional values among nursing students include social and personal values, which are established through clinical training and education [21]. That is, it is necessary for nursing instructors to consider education based on the ethical values in the curriculum and to provide value-based care for patients in the clinical environment in order to improve nursing professional values [33]. Therefore, when engaging with clinical training and curricula, students must continually strive to establish and maintain appropriate nursing values and an accurate perception of the nature of nursing. Since nursing professional values are essential qualities that must be established among nursing students, faculties should continue developing these values to ensure adequate PCC as clinical nurses.

However, perception of the nursing profession was not found to be a significant influencing factor on PCC competence. In the present study, the average score for the perception of the nursing profession was 3.99 ± 0.50, higher than the score of 3.65 ± 0.46 reported in a previous study [28]. This study was conducted among third- and fourth-year students who have experienced clinical practice, while previous research (Cho and Kim [28] was conducted among first- to fourth-year students. The average score for the perception of the nursing profession in this study may be unusually high, as this perception tends to become more negative after the clinical training experience in the third and fourth years [28]. Thus, the reason that the perception of the nursing profession did not affect PCC competence can be attributed to its high average score. Although the perception of the nursing profession did not affect PCC competence in our study, it is vital for nursing students. It is necessary to confirm the effect of the perception of the nursing profession on PCC competence through repeated research in the future, targeting nursing students from various nursing colleges.

In addition, satisfaction with major was not found as an influencing factor on PCC competence. The present study indicates the mean of satisfaction with one’s major as 4.10 ± 0.51, which was higher than 3.82 ± 0.06 and 3.87 ± 0.59, as reported by Park and Oh [25] and Woo and Park [24], respectively. This research found high levels of satisfaction with one’s major, which means that nursing students have a good understanding of their major and have a positive perception of the nursing profession [25]. Promoting nursing students’ satisfaction with their major could further consolidate their nursing professional values, thereby improving their caring behaviors [24,34]. Thus, educational strategies are needed to identify students with low satisfaction with their college major and address their issues. Further studies need to identify the factors influencing satisfaction with the major on PCC competence in nursing students.

This study is meaningful because it identifies nursing professional values as significant factors affecting PCC competence among nursing students, thereby underlining the need for strategies to foster nursing professional values among such students. Moreover, this study is the first to confirm, among nursing students undergoing clinical training, whether nursing professional values, satisfaction with major, and perception of the nursing profession are factors influencing PCC competence.

This study, however, has several limitations. It should be noted that, as our study featured an arbitrary convenience sampling approach conducted in three universities in Korea, a lack of representation among the findings limits their generalizability. In addition, since most of the tools of this study used the instruments developed in Korea, there is limited scope for replication studies worldwide. A few suggestions are made based on the study results. First, while developing programs for promoting PCC competence among nursing students, educators should consider strategies to improve students’ nursing professional values. Second, we suggest a study to identify PCC competence changes after providing these programs. Additionally, further research should be conducted to analyze nursing professional values, satisfaction with major, and perception of the nursing profession function as moderating variables or parameters concerning the PCC competence of nursing students.

## 5. Conclusions

For improving nursing students’ PCC competence, this study identifies the effects of nursing professional values, satisfaction with major, and perspective of the nursing profession on the PCC competence of nursing students. Nursing professional values (*p* < 0.001) were identified as a factor influencing PCC competence in nursing students undergoing clinical training (Adjusted R^2^ = 0.244). However, perception of the nursing profession, and satisfaction with major were not found to be significant influencing factors on PCC competence. This finding demonstrates that nursing professional values should be considered when developing educational programs to promote PCC competence among nursing students. Future studies are required to develop educational programs that include improving nursing professional values, and to examine its effectiveness.

## Figures and Tables

**Table 1 medicina-57-01295-t001:** Difference in the person-centered care competence by general characteristics (*n* = 158).

Characteristics	Categories	*n* (%)	Mean ± Standard Deviation	*t* or F	*p*-Value
Age (years)	<25	114 (72.2%)	3.64 ± 0.44	−2.09	0.038
	≥25	44 (27.8%)	3.80 ± 0.50		
Sex	Female	138 (87.3%)	3.66 ± 0.45	1.87	0.065
	Male	20 (12.7%)	3.86 ± 0.49		
Grade	Third-year	87 (55.1%)	3.68 ± 0.46	−0.16	0.874
	Fourth-year	71 (44.9%)	3.69 ± 0.47		
Religion	Yes	75 (47.5%)	3.74 ± 0.50	1.53	0.129
	No	83 (52.8%)	3.63 ± 0.43		
Satisfaction of clinical practice	Satisfied	105 (66.5%)	3.73 ± 0.46	2.10	0.125
	Neutral	46 (29.1%)	3.57 ± 0.47		
	Dissatisfied	7 (4.4%)	3.74 ± 0.36		
Academic score in the preceding semester	<3.0	14 (8.9%)	3.56 ± 0.38	0.77	0.464
	3.0~3.9	95 (60.1%)	3.72 ± 0.45		
	≥4.0	49 (31%)	3.67 ± 0.50		

**Table 2 medicina-57-01295-t002:** Mean score of person-centered care competence, nursing professional values, satisfaction with major, and perception of the nursing profession (*n* = 158).

Variables	Mean ± Standard Deviation	Range	Actual Range	
			Minimum	Maximum
Person-centered care competence	3.69 ± 0.46	1~5	2.7	4.9
Nursing professional values	3.92 ± 0.40	1~5	2.8	4.8
Satisfaction with major	4.10 ± 0.51	1~5	3.1	5.0
Perception of the nursing profession	3.99 ± 0.50	1~5	2.7	5.0

**Table 3 medicina-57-01295-t003:** Factors influencing nursing students’ person-centered care competence (*n* = 158).

Variables	B	Standard Error	*β*	*t*	*p*-Value
(Constant)	1.43	0.31		4.65	<0.001
Nursing professional values	0.57	0.08	0.49	7.18	<0.001
R^2^ = 0.249, Adjusted R^2^ = 0.244, F = 51.61 (*p* < 0.001)					

## Data Availability

The study data are available on request from the corresponding author.

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
