# Peer review of "Influencing Factors on Person-Centered Care Competence among Nursing Students Experienced Clinical Training"

_medicina, 2021, doi:10.3390/medicina57121295_

Round 1
Reviewer 1 Report
General comments:
The manuscript of ISSN 1648-9144 by the author and co-worker, evaluated the influencing factors on person-centered care competence by incorporating new factors from nursing students and their educations. The report about the importance of nursing professional training and educations provided new data foundations and novel visions in the field of nursing care.
Specific comments:
Line 36, the difference and similarity between person-centered and patient-centered care should be briefly introduced here.
Line 99, how were the insincere answers identified, any standards used?
Line 101, the instruments and methods used were confined to most like Korean or Asian. Could the instruments cited be more diverse worldwide?
Line 230, Currently, how do the faculties provide nursing students the improvements of their professional values? Please be more specific to describe what should be improved by the faculties.
Author Response
We appreciate your thoughtful comment here.
We have revised this in accordance with your recommendation.
We have attached the file.

Reviewer 2 Report
Dear Authors,
Many thanks for your manuscript submission to MDPI Journal of Medicina. Your research article investigates the effects of nursing professional values, the degree of satisfaction towards one’s major at college, and perspective of the nursing profession on the PCC competence of their corresponding students. While this short survey paper contains some merits, it may require one round of major revision. The major and minor issues as suggested to improve quality of this paper, are listed as below:
Major problematic issues to be addressed:
a) The keynote problem is person centered care (PCC), the authors may need to propose a detailed workflow on their approach to carry out PCC, and explain why it is important and how your approach outperform the related state-of-the arts. The current organization of Sections 1-2 are a bit too generic. Please consider improving the related parts of both sections.
b) Observing from the data collection and statistical analysis, I think the descriptions of their software tools and technical methods are vague. Also, given the limited number of data samples, the authors need to explain how does that make sense to represent the overall means for PCC competence.
c) Given the results in Table 1, the mean of all numerical values are within the range of 60~65, I think the authors need to specify any obvious difference between gender, age group, GPA, etc. Please update the related statements along with improving the experimental design.
d) This research article missed any useful figures to help potential readers understanding on PCC competence with respect to different nursing majors. The topic on study is hardly understandable; hence, I suggest the authors to be more concrete on their design of study (on clinical training) and adding proper workflow on the architecture of their procedure of survey work.
e) While the discussions are sufficient, the conclusion section is also a bit too generic, it should include useful information on opening questions to be solved and planned future work; meanwhile, it lacks keynote quantitative results as concluding remark; either problem exists in the Abstract session.
Minor issues suggested to be fixed:
a) Shorten the Abstract session within 150 words while including keynote quantitative conclusions.
b) Add a short paragraph on the main contributions of your work in the Introduction section, and edit the last paragraph on the organization of this paper (for each section).
c) Some notations should be italic, i.e., "K" in Line 136, and "n" in Line 172.
d) Parallel comparsion to other state-of-the arts should be included.
e) References: the style of non-italic / italic should be calibrated when citing the volume of the published article per journal; meanwile, some more recent works for PCC competence published in Years 2018-2021 should be supplemented; besides, I think the capital letter and linspacing issues should be fixed (multiple linespacing x 0.95) if the updated version were accepted for later publishing.
f) While use of English is acceptable, there are some grammatical issues to be calibrated in the context, and I believe some of the phrases and sentences, can be replaced with better statements. Please cooperate in the revising process.
We sincerely wish you the best of luck for paper editing. Stay well and thanks again for your interests on publishing at Medicina.
Best wishes,
Yours sincerely,
Author Response

(The authors gave the same response as above.)

Reviewer 3 Report
This paper collected survey data from nursing students in Korea to identify the influential factors on their person-centered care competence.
- Please cite the G*power 3.19 program in Line 95 and SPSS in Line 155.
- Please explain Cronbach's alpha and what the values mean in Line 110, 118, 124, and etc.
- In Line 167, the percentage of people who reported no religion is not consistent with what is listed in the table (52.5% vs. 52.8%).
- In Line 98, it is said that all 170 questionnaires were collected. It would be interesting to analyze the difference between the people who became lost to follow-up and those who completed the survey and are remained in the anlayses.
- In Line 193, the authors mentioned removing satisfaction with the major to avoid multicollinearity. However, it is not clear why there are only three variables remained in the final regression model shown in Table 3. Was any stepwise regression performed?
- It would be interesting to examine the univariate relationship between each predictor and the outcome variable in addition to the multivariate analysis alone.
- Based on the significance found in the regression models, it doesn't mean that correlation is causality. In that case, please try to avoid using the causal language including "influence" (Line 218, 221 and etc), "lead to" (Line 222), "effect on" (line 232), "improving" (line 259) and etc.
- Another limitation of the study is that no causal relationship is concluded from the analysis. Could you also elaborate on this point?
Author Response

(The authors gave the same response as above.)

Round 2
Reviewer 2 Report
Dear Authors,
Firstly, thanks so much for your carefully prepared review report and the updated research article. The authors have addressed almost all the reviewers' comments, and after the second round of review, I think it should qualify final acceptance. This version contains a few minor issues to be fixed, which I enumerate each of them (may not limit to these) as below:
a) The Introduction: A common way to summarize this section is to prepare a main summary of major contributions (the main contributions of this review is in three manifolds: i) ...; ii) ...; iii) ...) along with a short summary on the remainder of this paper (i.e., the rest of this paper is organized as follows ...) in the last two paragraphs, which are more clear than the current version.
b) In Line 96 of Page 2, please fix the "K Province" as "Kangwon Province". In Line 111 of Page 3, you may need to specify the scale of GPA (4.3 or 5.0 scale). In Line 168 of Page 4, I think the "t" of "t-test" might need to be italic or replace it with a capitalized "T". Please double check these trivial issues.
c) Figures and Tables: please rearrange the locations of Table 1 and Table 2 in Pages 4-6, which should not cross over two adjacent pages.
d) Conclusion Section: there are still some room for improvement on this short paragraph. The last sentence seems a bit generic, which can be expanded to a short paragraph on future work (orientations of prospective study with respect to the challenging issues to be solved), and independent from the main summary of work in concluding remarks. Thanks a lot!
e) References: [20], [22], [31], [32] cite unpublished Master's Thesis, it may follow the required format as MDPI template specified: [X] Author 1, A.B. Title of Thesis. Level of Thesis, Degree-Granting University, Location of University, Date of Completion.
[11], [25], [33] cite published journal work, where the abbreviation style within the name of each journal may require some minor updates as follows: [XX] Author 1, A.B.; Author 2, C.D. Title of the article. Abbreviated Journal Name Year, Volume, page range.
f) I have read through this research article and assert that the quality of presentation has been improvement, which some conjunctions are still hard to be readable. Hence, I recommend the co-authors to polish this version in more coherent ways and pay double-attentions to the transition words.
Once again, thanks so much for your careful edits. We wish you great success on further work towards paper acceptance. Stay safe!
Best wishes,
Yours sincerely,
Author Response
We greatly appreciate your thoughtful comments and have tried to incorporate your suggestions to the fullest extent possible. In addition, we have checked our manuscript many times and revised minor errors. Please find our responses below. We have highlighted revised parts in the manuscript.
